

# Sensitivity of local air quality to the interplay between small- and large-scale circulations: a Large Eddy Simulation study

Tobias Wolf-Grosse[1], Igor Esau[1], Joachim Reuder[2]

[1]Nansen Environmental and Remote Sensing Center, Bergen, NO-5006, Norway
[2]Geophysical Institute, University of Bergen, Bergen, NO-5007, Norway

*Correspondence to*: Tobias Wolf-Grosse (tobias.wolf@nersc.no)

**Abstract.** We present an analysis of the interaction between a topographically forced recirculation of the large-scale flow above an urbanized coastal valley and a local breeze-like circulation. We found that such an interaction can enhance the stagnation inside the valley under cold air pool conditions. Analysis of a large dataset of air quality measurements in Bergen, Norway, revealed that the most extreme cases of recurring winter-time air pollution episodes are usually accompanied by an increased wind speed above the valley. The 10 m ERA-Interim wind-speed distribution against local $NO_2$ measurements had a maximum at $3\ m\ s^{-1}$ in contrast to a monotonic decrease, as it would be expected from theory developed for flat, homogenous surfaces. We conducted a set of 16 Large Eddy Simulation (LES) experiments with the PALM model to account for the realistic orography of the mountains surrounding the city. The simulations were driven by the typical circulation above the valley during observed air pollution episodes, and a heterogeneous combination of constant temperatures over water and a constant negative sensible surface heat flux over land. The LES revealed a strong steering of the local circulation during cold air pool conditions by a land-breeze between the warm sea and the cold land. This breeze circulation is counteracted by a recirculation of the flow above the valley. For certain combinations of both, this leads to a maximum in the local stagnation. Furthermore, a relatively small local water body acted as a barrier for the dispersion of air pollutants along the valley bottom, dispersing them vertically and hence diluting them. These findings have important implications for the air quality predictions over urban areas. Any prediction not resolving these, or similar local dynamic features, might not be able to correctly simulate the dispersion of pollutants in cities.

## 1 Introduction

Urban air pollution is a major concern for urban dwellers and planners (Baklanov et al., 2007). Typically, urban air pollution is thought of in connection with large industrial areas and megacities with high local emissions (e.g. *Zhang et al.*, 2012). Air pollution episodes can, however, also affect smaller cities (e.g. *Junk et al.*, 2003; *Schicker and Seibert*, 2009; *Grange et al.*, 2013). In areas with comparatively low emissions, slow removal and long accumulation of locally emitted pollutants is usually responsible for the occurrence of air pollution episodes. For cities in mountainous areas, this can be caused either by





a local circulation trapping the pollutants or by local stagnation within the surrounding mountains (Rotach et al., 2004; Steyn et al., 2013). The most prominent example of the local stagnation is the frequently observed stably stratified Atmospheric Boundary Layer (ABL) in mountain basins, also referred to as cold air pools (e.g. *Reeves and Stensrud*, 2009; *Hoch et al.*, 2011; *Sheridan et al.*, 2014; *Hughes et al.*, 2015).

The same conditions, however, that lead to this high accumulation of pollutants, also make them notoriously difficult to study. The coupling of the local circulation inside a valley with the flow above can be complex and non-linear (e.g. Whiteman and Doran, 1993; Zängl, 2003). Additionally, an interaction with the local cold air pools can be further complicated by other local dynamic effects, such as breeze circulations (e.g. *Lareau et al.*, 2013). Numerical Weather Prediction (NWP) models often struggle with the correct prediction of air pollution episodes, in particular during stable

stratification and within mountainous terrain (e.g. *Berge et al.*, 2002; *Fay and Neunhäuserer*, 2006; *Baklanov et al.*, 2011). This has been associated with both, an insufficient spatial model resolution in order to resolve the relevant topographic features and problems with the parameterisation of the stably stratified ABL (Atlaskin and Vihma, 2012; Fernando and Weil, 2010; Zilitinkevich et al., 2015). This attribution (e.g. *Wolf et al.*, 2014) motivated the Pan-Eurasian Experiment community to include the air pollution transport and dispersion in the stably stratified boundary layers as one of the research priorities in

the collaborative cross-disciplinary research plan (Lappalainen et al., 2016).

Large eddy simulations (LES) can resolve much of the relevant turbulence dynamics. The computational costs of high resolution LES over larger urban areas are, however, too high to be used operationally. While this might change in the near future (Schalkwijk et al., 2015), LES models are today mostly used for the simulation of specific scenarios or for process studies in order to gain a deeper understanding of the flow features at hand (Bergot et al., 2015; Esau, 2012, 2007; Patnaik

and Boris, 2010). Increased local knowledge and general understanding of the relevant processes (e.g. Glazunov et al., 2016) can then help the forecasters to improve air quality predictions based on NWP models, as suggested by *Steyn et al.* (2013).

LES models are more and more used for the study of the air flow in model domains with realistic urban topography. To name a few, *Letzel et al.* (2008) simulated the flow on the neighbourhood-scale in Tokyo, Japan. *Esau* (2012) studied the dispersion of a passive tracer over central Paris, France. *Keck et al.* (2014) used an LES model for an assessment of the

effect of a new artificial island, densely covered by buildings, on the street-level ventilation in Macau, China and *Park et al.* (2015a, 2015b) studied the street-level ventilation in Seoul, South-Korea. Most studies, however, are conducted under neutral or slightly convective conditions. While the use of LES for the simulation of stably stratified conditions has become more common in the recent years, only few studies exist for the application of LES to the stably stratified ABL over urban and urban like areas. *Cheng et al.* (2010) and *Tomas et al.* (2016) for example studied the flow over arrays of surface

mounted boxes. *Xie et al.* (2013) studied the effect of a stably stratified approach flow on the dispersion conditions over a $1.2 \times 0.8 \, km$ large domain over central London. *Chen et al.* (2011) reported on a modelling system coupling WRF with an LES model. *Wyszogrodzki et al.* (2012) applied this modelling system for LES over a $1.6 \times 1.6 \, km$ large domain for convective and weakly stably stratified flows over Oklahoma City. A review of modelling of the urban ABL in a number of





different settings was conducted by *Barlow* (2014). To our knowledge there is up to now no high resolution study devoted to the simulation of the stably stratified urban ABL beyond the neighbourhood-scale.

In this paper, we will present LES of the topography and breeze induced circulation for the realistic topography of Bergen, Norway. Strongly stable stratification of the lowermost atmospheric layers is recognized as a precursor for the accumulation

of air pollutants and subsequent air quality hazards in urbanized valleys. The city centre of Bergen is embedded in a coastal valley ending at a large sea inlet, the Bergen fjord. Periods of high air pollution are observed during persistent temperature inversions, indicating the stably stratified conditions inside the valley (Wolf et al., 2014, hereafter WE14; Wolf and Esau, 2014). The observations of the typical circulation above the valley and the conditions inside it are used to drive the LES runs. We would like to focus on the interaction between the large-scale circulation above the valley, modified by the local

topography, and a local land-breeze between the cold land and the warm fjord under inversion conditions. This interaction is leading to the strongest stagnation and hence accumulation of pollutants inside the valley. The simulations should further serve as a proof of concept for the added value of high resolution LES for the analysis of urban air pollution dispersion in complex mountain terrain.

This study is structured as follows. In section 2 we will present an observational dataset that motivated this work. In the

third section we will outline the applied methodology and the results will be presented in the third, and summarized and discussed in the last section.

## 2 Observational perspective on air pollution in coastal cities

For coastal cities land/sea breeze circulations can have a strong impact on the local circulation. This can lead to a transport of pollutants from high emission areas onwards to inhabited areas (Gariazzo et al., 2007) or a closed recirculation and hence

accumulation of pollutants (Crosman and Horel, 2016; Lo et al., 2006; Rimetz-Planchon et al., 2008). Most of these works have in common that the background circulation reduces the relevance of the local effects (Crosman and Horel, 2010). For coastal mountain cities there is, however, also the possibility of an interaction between local circulations such as local breezes and cold air pools (Holmer et al., 1999; Lareau et al., 2013). These local circulations can be either dominating the local circulation forced by the large-scale flow or interact with it. Both can lead to a deviation from the usual situation of a

weaker background flow leading to higher local accumulation of air pollutants.

Fig. 1 illustrates this based on the distribution of the $NO_2$ concentrations against wind speeds above and inside the Bergen valley. Wind speeds inside the valley are measured on a mast on top of the Geophysical Institute (GFI, compare Section 3.1) at $12\,m$ above the roof (approximately $50\,m$ height). The air flow at this height is unobstructed by other buildings. The measurements should therefore be representative of the air flow in the middle of the valley. Quality controlled hourly data

for the entire period between 2003 and 2013 were readily available online (Norwegian Meteorological Institute, 2016). The expected logarithmic decrease of the $NO_2$ concentrations with increasing wind speeds inside the valley is visible for both an urban background (UB) and high pollution (HP) reference station (data downloaded from Norwegian Institute for Air



Research, 2016). For the surface ($10\,m$) wind speeds from the ERA-Interim (EraI) reanalysis product, however, such a clear decrease does not exist. Due to the low resolution of EraI, the $10\,m$ winds are not influenced by small-scale topographic features such as the Bergen valley. They should therefore represent the large-scale flow that is only modified by the Norwegian topographic features on scales larger than a few tens of kilometres. For cases of high $NO_2$ air pollution there is a

maximum for wind speeds around $3\,m\,s^{-1}$. This maximum in the concentrations at some nonzero background wind speed suggests an interaction of the background flow with a local forcing under the cold air pool conditions. Some combination of all three circulation features then leads to a maximum stagnation over the valley bottom.

Thus, we observe that any air quality prediction based on meteorological fields from models that are not resolving this local forcing could fail to resolve the highest local air pollution concentrations. We assume that the most relevant local forcing in

the Bergen valley is the breeze circulation caused by the temperature difference through the land-sea interface. The relevance of the interplay between the larger scale circulation, the local topography and the local forcing is therefore assessed in this study.

## 3 Model experiments

### 3.1. Geographical description of the simulation domain

Bergen, Norway is located at the Norwegian West coast (60.4° N, 5.3° E). The central part of the city is located in a curved valley with a minimum width at the valley base of approximately $1\,km$ (Fig. 2). The surrounding mountain tops are between 344 and $642\,m$ high. Inside the valley there are a number of water bodies aside the fjord. Only the largest ones close to the city centre are explicitly treated in this study.

Since 2003 two measurement stations have been monitoring the street level ($2\,m$ above surface) air quality in the Bergen

valley, a high pollution reference station (HP) that is located nearby one of the busiest crossroads in the city and an urban background reference station (UB). The valley favours frequent winter-time ground-based temperature inversions (WE14) leading to the exceedance of air quality thresholds for $NO_2$ and $PM_{2.5}$ especially in areas with high traffic. In contrast to some cold air pools in large valleys (e.g. *Zhong et al.*, 2001), the cold air pools connected to air quality hazards in the Bergen valley are caused by ground-based temperature inversions. They cannot exist without persistent LW radiative cooling such

as during fair-weather events in wintertime with no or only little solar insolation and clouds. Temperature inversions in the Bergen valley usually appear in connection with a specific circulation pattern both in- and outside the valley. WE14 analysed the $10\,min$ wind-measurements from an automatic meteorological station (AMS) on top of GFI together with measurements of the vertical temperature profile above GFI. They found almost exclusively down-valley winds during measurements with temperature inversions, while during measurements without temperature inversions both up-valley and down-valley

channelled flows existed. This, together with the previously mentioned south-easterly background flow above the Bergen valley during high air pollution events, gave reason to assume that the preferred wind direction inside the valley simply follows the large-scale wind direction above. In this case the mountain to the south-east of the Bergen valley would shelter



the valley. This is, however, a too simplistic picture since the sheltering mountains would represent a backward facing step for the approaching winds that can lead to very complicated flow regimes (Mohamad and Viskanta, 1995).

During persistent winter-time temperature inversions in the Bergen valley the local lakes often freeze over. The fjord, however, remains mostly ice free, causing large temperature contrasts between the relatively warm fjord surface and the cold

land surface possibly leading to the local breeze circulation inside the valley. The large water body roughly in the middle of the domain is connected to the fjord only though a narrow channel. It is therefore rather brackish water and tends to freeze over with a thin layer of ice during persistent temperature inversion episodes. It is consequently treated as a lake later on.

### 3.2. The model

In the experiments for this study we used the Parallelised Large-Eddy Simulation Model for Atmospheric and Oceanic flows

PALM (Maronga et al., 2015; Raasch and Schröter, 2001). PALM solves the finite difference numerical realisation of the non-hydrostatic, filtered, incompressible Navier-Stokes equations in their Boussinesq approximated form. The model employs a 1.5 order closure using a subgrid-scale TKE balance equation (Deardorff, 1980; Moeng and Wyngaard, 1988; Saiki et al., 2000). Advection terms are computed using the 5th order scheme after Wicker and Skamarock (Wicker and Skamarock, 2002). The model time-step is adjusted dynamically. The incompressibility condition is satisfied with a

predictor-corrector method, using the Temperton fast Fourier transformation to solve the Poisson pressure equation. Spatial discretisation is done on an Arakawa type C grid. Topographic features of the urban area are simplified as ground-mounted boxes. Topographic input needs to be specified as a separate input file that fits the horizontal model grid. Vertical discretisation of the surface height onto the model grid is done automatically by the model.

### 3.3. Modifications to the PALM code

PALM runs either with the Neumann surface temperature boundary condition (BC), where the kinematic surface heat flux ($H_s$) is prescribed, or with the Dirichlet BC, where the surface temperature ($T_s$) is fixed. In the case of the Dirichlet BC, heat fluxes on horizontal surfaces are calculated by assuming a Prandtl layer. In order to be able to simulate the potential breeze effect in the valley, we added the possibility to use mixed Dirichlet and Neumann BC to the model. Consistent with the large heat capacity of water, we kept $T_s$ constant over the water-covered parts of the computational domain. This allows for the

development of the typical pattern of organised turbulence over the water surface, as it has been repeatedly found for breeze circulations, e.g. induced by arctic leads (Esau, 2007; Lüpkes et al., 2008). Over the land surface we kept $H_s$ constant in order to represent the effect of LW radiative cooling. While specifying negative $H_s$ can be problematic for LES studies (Basu et al., 2008), it was necessary here to incorporate the inhomogeneities in surface temperature due to the large differences in surface elevation and the land-sea interface.

The heterogeneous fields for $H_s$ and $T_s$ are specified via separate ASCII files, containing arrays with the same dimension as the computational domain (*Maronga and Raasch* (2013), Björn Maronga personal communication). The files are read into the model in the beginning of each simulation. The arrays contain the value 1 if a specific heat flux or surface temperature





should be used for the corresponding model grid-cell and 0 otherwise. The values of $H_s$ and $T_s$ for all grid-cells are then prescribed for a freely chosen number of times during the run via a separately read ASCII list. The use of mixed BC is reached by running the model with the Dirichlet BC and correcting $H_s$ back to the prescribed value over the land surface area for every time step, including a correction of the calculation of the friction velocity and temperature in the Prandtl layer

routine. At each grid-point, either $H_s$ or $T_s$ have to be prescribed. In order to be able to study the dispersion of pollutants from different emission sources, we used the same approach for reading heterogeneous fields for the surface flux of a passive tracer $F_s$. To avoid unphysical recycling of the passive tracer due to periodic boundary conditions, we set the passive tracer to 0 at the lateral boundaries of the computational domain.

### 3.4. Domain

The simulations are done for the realistic topography of the city of Bergen, Norway. For this, we received laser scanning data from the municipality of Bergen for a $5\,km$ square around Bergen city hall. In the choice of the final domain size, we tried to include the mountains directly surrounding the inner part of the city, while attempting to keep the computational domain as small as possible. At the lateral boundaries, we used periodic BC. The Bergen valley is open towards the north-west and south-west. In order to allow for a more realistic free flow along the valley axis, we created an artificial channel at

the northern end of the domain. While necessary in order to avoid an unnatural stagnation in the southern part of the valley, this channel might also alter the circulation over the fjord at the northern boundary of the computational domain. For future studies this should be avoided by using a larger north-south extent of the domain. At the lateral boundaries we used a 1000 m wide buffer zone, both in the x- and y- directions. In this buffer zone surface elevations are linearly interpolated in order to guarantee a smooth transition through the periodic boundaries.

Polygons of all water bodies in Bergen were provided by Bergen municipality. From this, we produced the input files for the areas with constant $H_s$ over land and constant $T_s$ for grid-boxes that were covered by more than 50 % of water. As a simplification we ignored most fresh-water lakes. The final domain consists of $1024 \times 1024$ grid nodes in the horizontal $x$- and $y$-directions including the buffer zones, and 128 levels in the vertical $z$-direction. The model resolution is 10 m for each coordinate axis in the lower $750\,m$ of the computational domain. Above $750\,m$ the grid is vertically stretched by 1 % for

each additional grid level. The total domain size is $L_x = L_y = 10240\,m$ and $L_z = 1451\,m$ – well above the highest mountain top at $650\,m$. We smoothed the topography with a running mean filter over three grid cells in both lateral directions.

The extent of the fjord in this setup is roughly $4\,km$ in the E-W direction and $3\,km$ in the N-S direction. The western boundary of the fjord is representative for the location of a large island closing the fjord, except for a $700\,m$ wide passage.

To the north, the fjord extends in reality much further than in our setup. The artificially set northern boundary therefore reduces the fjords extend. A comparison to *Esau* (2007) shows, however, that $2 - 4\,km$ is the size where the drag of air



from warm arctic leads stagnates at its maximum, meaning that the convergence in the N-S direction should be sufficient to cause a realistic drag of air in the valley.

It should be noted that a $10\,m$ resolution is clearly too coarse in order to resolve the circulation in street canyons (Letzel et al., 2008, 2012). It is also too coarse for the simulation of the stably stratified ABL over flat and homogeneous surfaces
(Beare et al., 2006; Mason and Derbyshire, 1990). The focus of this study, however, is on the effect of the larger topographic features (valley width around 1 km) and the convection over the fjord on the circulation within the valley. Both forcings should be sufficiently reproduced with the chosen resolution.

### 3.5. Numerical Experiments

The numerical experiments are listed in Table 1. All experiments share a common basic setup: The latitude was $60.38°\,N$,
corresponding to Bergen. At initialisation, the surface potential temperature was $273.15\,K$ with constant potential temperature up to $650\,m$, and a constant potential temperature gradient of $5.5 \times 10^{-3}\,K\,m^{-1}$ above. This is the mean potential temperature gradient above Bergen during high pollution cases derived from the EraI data set. $H_s$ over land was chosen as $-0.025\,K\,m\,s^{-1}$, corresponding to approximately $-25\,Wm^{-2}$. This is in agreement with heat fluxes found from observational studies (Brümmer and Schultze, 2015; Nordbo et al., 2012). In addition to the cases with mixed BC, we also
simulated a test-case with a constant $H_s = 0\,K\,m\,s^{-1}$ over the entire domain, representing a situation with neutral stratification.

All simulations were run for $12\,h$ in order to reach a quasi-equilibrium state. During this time we kept the surface potential temperature over most water bodies at $273.15\,K$. The surface potential temperature of the elongated lake in the north-east of the Bergen valley was $273.89\,K$. This is the potential temperature corresponding to a temperature of $0\,℃$ at the height of the
lake surface (approximately $75\,m$). The temperature chosen over the fjord is given in Table 1 for each simulation. For cases 1 trough 12 we included emission of a passive tracer over the entire urban area into the model simulations. As emission rate we chose an arbitrary value of $1\,[kg\,m^{-2}s^{-1}]$. In order to mimic the actual extend of the build-up city area we only allowed for emissions over land-covered grid-cells with surface elevation below $70\,m$ in the original input file. For the last three cases we used the same emission strength, but only for the area covered by the largest street in the valley (see Fig. 1). The
exact geographic location of this street was provided to us by Bergen municipality.

PALM simulations are usually driven with an imposed geostrophic wind profile. For the geostrophic winds in our model experiments we used three different scenarios illustrated in Fig. 2. The profile with the lowest wind speeds follows the mean of the wind speed profiles from EraI above Bergen during days with high $NO_2$ air pollution (at least one hourly mean measurement exceeding $200\,\mu g\,m^{-3}$ at the high traffic reference station). Because of the varying height of the EraI model
levels, we linearly interpolated the wind profiles between the nearest model levels to $410, 450, 600, 800, 1000, 1200$ and $2000\,m$ height before averaging. Since EraI has a rather low resolution, the lowest grid-layer over Bergen is at approximately $410\,m$, depending on surface pressure. In four out of the total 45 high pollution days in the measurement




record, the lowest model layer was centred above $410\ m$. For these cases, no wind speed was available at the $410\ m$ height. As the lowest point in the PALM domain is the sea surface, it was, however, necessary to specify a wind speed below $410\ m$ height. We chose to use the wind speed from 410 m in EraI at $100\ m$ height in our PALM simulations and $0\ m\ s^{-1}$ wind speed at sea level. The mean wind direction profile during all high pollution cases changed from 100° at the lowest levels to

120° higher up. For simplicity, we kept the wind direction constant at 110°. The two higher wind speed scenarios follow vertical wind speed gradients of 1.5 and 2 times the mean gradient for all height levels above 100 m.

### 3.6. ERA-Interim data

EraI data are available from the ECMWF archive (Dee et al., 2011; ECMWF, 2016). The resolution of EraI (T255) is too coarse to resolve any of the relevant features of the Bergen valley. We therefore used the EraI wind speed and direction for

the specification of the background winds in PALM. We downloaded data at a horizontally interpolated grid of $0.25^{\circ}$ resolution and used the mean over the two grid-boxes centred at $5.5°\ E$, $60.25°\ N$ and $5.5°\ E$, $60.5°\ N$ that represented best the location of Bergen. For calculating the daily mean fields, we used EraI data at a 3-hourly resolution from a combination of the analysis steps at $6, 12, 18$ and $0\ UTC$ plus the 3 and $9\ h$ lead time forecasts of the 0 and $12\ UTC$ analyses.

### 4. Results

### 4.1. Main features of the Bergen valley circulation

The setup used here was chosen to study the potential effect of the breeze-induced circulation on the dispersion of pollutants in the Bergen valley under the conditions of typical winter time temperature inversions. Here we will highlight the most relevant features and briefly compare them to the results of the observational study conducted by WE14 in order to better understand the circulation in the valley. By this we will also investigate the potential and limitations of the chosen setup for

the proposed flow interactions. We use case 3 as a baseline that uses the mean wind speed profile during all high air pollution cases. The fjord temperature of 2.5 ℃ should be realistic for typical persistent winter-time temperature inversions. After 12 hours all simulations are in quasi-steady-state conditions. The resulting $2\ m$ temperatures over the fjord and area 1 and 2 of the city are given in Table 1. While the absolute temperatures over the different areas are irrelevant for this study, their differences are an indicator for the breeze circulation. By design, the difference between the air temperatures over the

fjord and the temperatures over the city are increasing with increasing fjord surface temperatures, consequently applying the forcing for a breeze circulation. In addition, the air temperature over the interior part of the valley is, except for case 12, lower than the air temperature over the outer part of the valley. This could exert another breeze-like forcing between these two parts of the valley. The dependence of the breeze effect on the prescribed fjord surface temperature decreases over time. This lets the different simulations, based on the same wind profile, converge. The reason is that over land the heat flux is

fixed, not the absolute temperature. The land surface temperature can therefore adapt to the fjord temperature due to the



advection of relatively warmer air. How fast this equilibrium is established depends on the total volume of the computational domain and the land and sea fractions. No convergence was visible for any of the wind speed scenarios simulated here.

Fig. 4 shows the wind field from case 3 on the vertical level centred at $55\ m$ together with $H_s$. This is the first vertical level above the AMS used in WE14 (marked in Fig. 2). The south-easterly, down-valley mean flow above GFI is reproduced in our simulation. The mean $H_s$ over all water bodies is $143\ Wm^{-2}$. Maximum values of up to $1000\ Wm^{-2}$ for a few grid nodes are located in direct proximity to the coast in areas with the strongest seaward flow, as a result from the temperature contrast between the land and the fjord surface. Over the fjord, $H_s$ is not simply decreasing towards the middle of the fjord, but reaches its minimum in the areas of flow convergence. These are for case 3 organised in form of two convergence lines.

Fig. 5 shows local profiles of temperature, wind speed and wind direction over and around the large water body in the middle of the valley. Area 2 in the plot is centred over GFI. The south-easterly flow is visible up to a height of $95\ m$. For areas 1 and 2 there is a gradual eastwards rotation of the wind direction, likely caused by a combination of the proximity to the warm sea inlet and the local topography. Between $95\ m$ and $105\ m$ height, the wind direction jumps from easterly to north-westerly before rotating back to mostly easterly wind between $300\ m$ and $400\ m$. Over area 3, the wind direction remains constant for the lowest parts of the ABL before gradually turning to the same north-westerly wind around $300\ m$.

Case 0, with missing surface temperature and heat flux heterogeneity, also showed a north-westerly up-valley flow, both above the fjord and inside the valley. However, the down-valley flow at the valley bottom was absent. The north-westerly flow seems to be a persistent feature of the Bergen valley topography under the given geostrophic wind profile, to some degree balancing the south-easterly flow that is caused by the convergence over the fjord. The flow in the valley is therefore not simply following the upper air wind-distribution, but clearly locally forced. Tests with slightly smaller and 1.4 times larger domain sizes under neutral conditions, showed a similar north-westerly flow. Due to a lack of laser scanning data over the larger domain, this test simulation is based on topographic input from a digital terrain model that does not include buildings and therefore underestimates the topographic height over the city. Due to missing information on the land-sea fraction for this data set, no test simulations were conducted with the larger domain size and non-neutral conditions.

The north-westerly flow higher up in the valley ABL is only visible for the simulations with 270.65, 273.15 and 275.65 $K$ fjord surface temperature, and case 11 with 278.15 $K$ fjord surface temperature and wind speed scenario 3. For the simulations with higher fjord surface temperatures it is not detectable and might be masked by the south-easterly flow from the convergence over the fjord. A reverse flow above a breeze circulation is usually associated with the returning branch of the breeze circulation. An increase in the strength of the land-breeze should therefore also result in an increase of the return branch. This being not the case for our simulations here for increasing fjord surface temperatures indicates that the north-westerly flow higher up in the valley ABL is not the return branch of the land-breeze above the valley bottom.

The modelled temperature profiles in Fig. 5 show inversions up to $135\ m$ height split into two separate inversions with two closely adjacent maxima for areas 1 and 2. The top of the lowest inversion was at $75\ m$ and the highest at $135\ m$. For area 3, there is only one inversion ending in between the two maxima from area 1 and 2. The results of the measurements with a passive microwave temperature profiler (MTP-5HE from Attex) on the rooftop platform of GFI, presented in WE14, provide





the possibility for a comparison with observed inversion heights in the Bergen valley. This instrument does not measure the vertical profiles directly above the instrument, but averages over variable horizontal distances based on the scanning angle. The measured temperature profiles are therefore rather comparable to a combination of the profiles along the horizontal measurement path with the lower height levels including a larger horizontal distance along the measurement direction than

5 the higher height levels. Areas 1-3 are roughly placed along the measurement path of the microwave radiometer, and should therefore be appropriate locations for a validation. Due to the limited vertical resolution, the angular scanning microwave radiometer smoothes out fine structures, such as e.g. two closely adjacent maxima. For the other simulated cases, the inversion heights were rather similar, varying between 85 and 145 $m$. The observed inversion depths in the Bergen valley are typically ranging between 70 and 270 $m$, with the majority of observations between 70 and 220 $m$. Inversion episodes

lasting longer than 2 hours were on average most frequently 170 $m$ deep. This indicates that our LES simulations somewhat underestimate the inversion depth. The modelled maximum inversion strength in the order of $1 - 3\,K$ is in accordance to the observations (WE14). Since the simulations presented here are rather idealised, it is likely that relevant physical processes for a more realistic representation of the inversions, such as large-scale subsidence or long-wave radiation divergence in the atmosphere, are not fully considered. (Hoch et al., 2011) suggested a cooling of the inversion top together with a

simultaneous heating of the air layers above and below. This could be a possible mechanism for further inversion growth. In addition, the temperature profile above the inversion is usually weakly stable, whereas our simulations show a well mixed profile almost to the top of the computational domain. This is caused by the application of the periodic boundary conditions in the model simulations. A nudging of the mean potential temperature gradient above the valley could solve this problem, but is not available in the model setup we used. Improving the representation of both processes in the model is expected to

result in a growth of the inversion depth. Furthermore, the flow along the valley bottom through the southern domain boundary might be overestimated in our simulations. Even though, this flow feature might to some extend also exist in reality, as there are large water bodies further south in the Bergen valley that should cause convergence and hence a draining of air out of the city centre, this could reduce the potential for cold air pooling, Finally, the relatively low spatial resolution for the simulation of the stably stratified ABL might also negatively impact the representation of the temperature inversions

in the Bergen valley.

However, our simulations produced ground-based inversions higher than the 55 $m$ model level. This means that they should be able to give us relevant information on the mean flow around the height of the AMS and below, the levels most relevant for the dispersion of locally emitted air pollutants.

### 4.2. The interplay between the local and the larger scale conditions

The range of selected fjord surface temperatures and geostrophic wind speeds allows to investigate the interplay between the south-easterly down-valley flow, triggered by the convection over the fjord, and the north-westerly up-valley flow, forced by the flow above the valley, and its effect on the dispersion of pollutants inside the valley. Fig. 6 shows the wind-field at 55 $m$ height and the terrain following concentration of the surface emitted passive tracer at 2 $m$ above the ground. The runs with





fjord temperatures of 2.5 ℃ and above show clear signs of flow convergence in the wind field over the fjord and a distinct signature of prevailing down-valley flow. For the leftmost panels with 0 ℃ fjord surface temperature, this convergence line is pushed all the way towards the coastline and the flow in the exterior part of the Bergen valley is even reversed towards an up-valley flow. For case 9, the convergence line is masked by this up-valley flow at the 55 $m$ height level and therefore only

5 visible for the lower level wind fields. The results show a gradual movement of the convergence line with decreasing fjord surface temperatures and increasing geostrophic wind speeds inwards towards the city centre. This is caused by a weakening of the convergence over the colder fjord and the overall flow pattern is more and more dominated by the up-valley recirculation, especially for the scenarios with higher wind speeds. It should, however, be mentioned that the outflow out of the artificially generated channel at the northern border of the computational domain seems to interact with the up-valley

flow, enhancing it and hence pushing the convergence line towards the land. Down-valley flows are almost never observed during temperature inversions in the valley, but they are visible in our simulations. While the coldest fjord surface temperatures considered here are rarely observed, possibly explaining this lack of observations of the up-valley flow, it could also be artificially enhanced in our simulations by this boundary effect. This is, however, a persistent feature of all simulations. It might therefore shift the balance between the up-valley circulation, forced by the flow above the valley, and

the breeze circulation towards lower geostrophic wind speeds and higher fjord temperatures, respectively, but is not expected to change the conclusions on the existence of the balance itself.

The effect of this on the circulation inside the valley is summarised in Fig. 7 in terms of the horizontal mean of the passive tracer concentration 2 $m$ above the ground and the wind speed and direction at the 55 $m$ height level and 10 $m$ above the ground. For each of the given wind speed scenarios, there is a combination of geostrophic winds and local breeze

circulations, that leads to a maximum in the stagnation in the exterior part of the valley (area 1, Fig. 2). For the winds, this is visible as a plateau for the wind speed in scenarios 2 and 3, and a turning of the wind direction both at the 55 $m$ height level and 10 $m$ above the ground. Based on this, it can be assumed that there would be a minimum in the local wind speeds at intermediate fjord temperatures between 0 ℃ and 2.5 ℃. For scenario 1, with the lowest considered wind speeds, this balancing is not yet reached for the 10 $m$ wind, while the wind at the 55 $m$ height level is already rotated towards an up-

valley flow. This indicates, however, that the rotation of the winds at 10 $m$ above the surface would occur at fjord surface temperatures below 0 ℃. To investigate this further we conducted a test-simulation with a fjord surface temperature of −2.5 ℃. This scenario, however, led to an unrealistic maximum in convergence over all other water bodies, except for the fjord. While the absolute temperatures of the fjord and land surface are not relevant, the relative temperature differences are. During the winter the fresh water bodies are usually colder than the fjord. The constant temperature of 0 ℃ over the fresh

water bodies in our simulations, therefore, causes an unrealistic circulation in the valley at fjord surface temperatures below 0 ℃.

For the interior part of the valley (area 2, Fig. 2) a balance, as for the exterior part of the valley, is not visible. The set of simulations with the lowest fjord surface temperatures of 0 ℃ show both, the lowest wind speeds and maximum pollutant





concentrations and no reversal of the wind direction is seen for any of the simulations at the height of the GFI AMS. One reason for this distinctly different behaviour of the interior part of the valley is the colder temperature there. The temperature difference between the $2\,m$ surface air temperatures over area 1 and 2 at least for the lower fjord surface temperatures is similar to the temperature difference between the surface air temperatures over area 3 and 1 (see Table 1 for comparison).

The different land surface temperatures therefore exert an additional forcing on the air in the interior part of the valley. Furthermore, as the large water body in the middle of the valley is warmer than the surrounding land, it causes another centre of flow convergence similar to the fjord. It enhances an up-valley flow over area 1, while it weakens it for area 2. This effect is increasing with decreasing fjord surface temperature due to the changes in land surface temperature between the different model simulations. Especially for the cases with $0\,°C$ fjord surface temperature, however, the convergence over the water

body in the middle of the valley is most likely overestimated, since the brackish water lake usually cools off much faster than the fjord, resulting in a smaller temperature difference between the lakes and the surrounding land. A thin layer of ice, as it is typical on this water body, utterly reduces this effect.

The impact of the wind circulation on the dispersion of the passive tracer is two-fold. Locally, more stagnant conditions in area 1 lead to an increased accumulation of pollutants in this region. The concentration over area 2 is in general higher than

over area 1, except for case 7. This part of the valley is more protected from the geostrophic winds than area 1. In addition, the wind direction plays a more important role for area 1. An up-valley flow automatically leads to a lower pollutant concentration there, since air from the fjord, with considerably lower tracer concentrations, will be transported inland.

### 4.3. Analysis of single road contribution

For many valleys the main pollutant emissions over land come from single transit roads. The same is the case for the $NO_2$ emissions in the Bergen valley. The main emission source for traffic emitted pollutants is the transit road marked in Fig. 1. Emissions from a reduced area give insight into the efficiency of the horizontal dispersion for the most relevant areas that was obscured by the areal emissions assessed before. Therefore, we repeated cases 9 to 11 with tracer emissions only from this road. The wind fields, together with the passive tracer concentrations $2\,m$ above the surface, are shown in Fig. 8. We

chose to use sc 3 for the wind speeds in order to see the full range of conditions, from a down-valley flow to maximum stagnation.

The up-valley flow in case 13 transports the passive tracer away from the city centre. As the emissions from the street here are only scaling with the area covered by the street, ignoring traffic density and pattern for the emission factors, the large interchange road is causing the highest density of emissions. However, the high tracer concentrations are not transported to

other places in the valley. They are rather caught by the convection from the lake in the middle of the valley, which serves as an effective barrier also for the tracer transport from this street. This is, to a reduced degree, also visible for the other two cases. Emissions from area 2 contribute less to the pollution directly north of the lake. This indicates that even relatively smaller interior water bodies could improve the urban ventilation through driving convective and local-scale circulations.





For the other two cases, the mean down-valley flow transports the tracer along the valley axis. While for case 14 there is still an accumulation visible because of a relative stagnation, the down-valley flow is sufficiently strong and keeps the passive tracer concentrations at a low level over large parts of the city centre for case 15.

## 5. Summary and Discussion

In this study, we run a set of large-eddy simulations with the PALM model to evaluate the role of local circulations and their sensitivity to the interplay between external large-scale forcing and local forcing due to heterogeneities in the surface conditions. Specifically, we addressed the sea-land temperature difference, the large-scale wind speed and the imposed static stability of the lower atmosphere for typical conditions leading to high air pollution in Bergen, Norway.

Urban settlements in mountainous terrain at high latitudes are especially prone to adverse effects of temperature inversions

on the air quality. A lack of solar heating in winter and topographic constraints on the low level atmospheric circulation can lead to an accumulation of air pollutants near the surface. At the same time, large sea-land surface temperature differences create local and meso-scale circulations in coastal cities, which can partially compensate or even overwhelm the low level circulations forced by the large-scale atmospheric flow. Whether and to what degree will the circulations due to the surface heterogeneity impact the urban ventilation and hence the air quality? The outcome depends not only on the case specific

geographical features of the terrain and the specific emissions in the city, but also on more universal physical mechanisms and scalings. Such dependencies have already been extensively studied for flat surfaces and more regular heterogeneities, e.g. related to the sea ice fractures or leads in the Arctic Ocean (Esau, 2007) or to idealized surface plot patterns (van Heerwaarden et al., 2014). Published studies on urbanized mountain valleys often did not have sufficiently fine resolution to reproduce interactions between the flow above and the local flow features inside the valley. Moreover, they have not

addressed these interactions in the stably stratified boundary layers, when the air pollution may be particularly dangerous.

The conducted joint analysis of the $NO_2$ concentration and meteorological parameters for the Bergen valley revealed an unexpected build-up of air pollution under synoptic situations with significant non-zero large-scale winds. There was a distinct maximum in the distribution of observed $NO_2$ concentrations against the Era-Interim surface wind speeds at around $3\ m\ s^{-1}$. This behaviour is inconsistent with the usually assumed monotonic concentration-wind speed dependencies

and the faster depletion of valley cold air pools with increasing wind speeds (e.g. *Zängl*, 2003; *Lareau and Horel*, 2014). The behaviour of monotonically decreasing concentrations against wind speeds is, however, recovered for the actually measured surface winds. This indicates that some sufficiently strong local circulations emerge near the surface that are able to counteract the large-scale winds, but are not resolved in ERA-Interim. We therefore studied the physical mechanisms, dynamics and sensitivities to the surface features for those circulations with a set of 16 PALM scenarios (Table 1) for the realistic terrain

surrounding the city.

The simulations showed that, both, the local circulation, forced by the large-scale flow, or the locally forced breeze circulation could dominate the dispersion of air pollution in the lower valley atmosphere. The maximum pollutant trapping




under prevailing inversion conditions was dependent on the exact interplay of the three circulation features, i.e. large-scale flow, topographic steering and breeze circulation. The simulations with the lowest fjord surface temperatures showed a mean up-valley flow dominated by the topographically steered recirculation of the large-scale flow. The pollution emitted from urban activities was represented by a passive tracer emitted from the land surface over the urban area. The up-valley winds

therefore caused the advection of tracer free air from over the fjord into the city centre. The simulations with the higher fjord surface temperatures showed a mean down-valley flow dominated by the breeze circulation. For the highest fjord surface temperatures this lead to an efficient depletion of the tracer emitted over the urban area. For the simulations with intermediate temperatures, however, both circulation features balanced each other, leading to local stagnation and an accumulation of the tracer.

Perhaps one of the most interesting implications of this study is the possibility to analyse pollution scenarios for a specific area induced by concrete emission sources. Such inverse diffusion problems are frequently solved through a Green function method for regular domains (Lin and Hildemann, 1996), but for irregular domains, the direct simulations remain more computationally efficient. The simulations demonstrated that the strongly localized concentrations are rather sensitive to small-scale convective sources such as interior lakes. These effectively work as barriers for the dispersion of pollutants near

the ground. An approach to predict local air quality without resolving such local features will not be able to simulate the pollutant dispersion pattern correctly.

## Acknowledgements

This study was funded by the GC Rieber foundation. We thank Bergen Municipality for provision of the topographic input data, the land-sea mask and the position of roads in the Bergen valley. We further thank the Norwegian Meteorological

Institute and the Geophysical Institute at the University of Bergen for providing wind measurement data, the European Centre for Medium-Range Weather Forecasts for the ERA-Interim data and the Norwegian Institute for Air Research for the air quality measurement data. Sigfried Raasch and Björn Maronga with the Institute of Meteorology and Climatology at the University of Hannover were helpful in adapting and applying the PALM code.

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



**Table 1: model simulations**

| Case | Wind scenario | $\theta_{fjord}$[K] | $T_{2m}(fjord)$ [K]** | $T_{2m}(area\ 1)$ [K]** | $T_{2m}(area\ 2)$ [K]** |
|---|---|---|---|---|---|
| Case 0* | sc 1 | - | - | - | - |
| Case 1 | sc 1 | 270.7 | 271.5 | 269.8 | 268.0 |
| Case 2 | sc 1 | 273.2 | 273.6 | 271.3 | 269.1 |
| Case 3 | sc 1 | 275.7 | 275.4 | 272.0 | 270.1 |
| Case 4 | sc 1 | 278.2 | 277.4 | 273.0 | 271.6 |
| Case 5 | sc 2 | 273.2 | 273.7 | 271.5 | 269.0 |
| Case 6 | sc 2 | 275.7 | 275.4 | 272.2 | 270.8 |
| Case 7 | sc 2 | 278.2 | 277.4 | 273.3 | 272.8 |
| Case 8 | sc 2 | 280.7 | 279.1 | 274.4 | 273.3 |
| Case 9 | sc 3 | 273.2 | 273.7 | 271.7 | 269.8 |
| Case 10 | sc 3 | 275.7 | 275.5 | 272.5 | 271.0 |
| Case 11 | sc 3 | 278.2 | 277.3 | 273.5 | 272.4 |
| Case12 | sc 3 | 280.7 | 279.1 | 275.1 | 275.2 |
| Case 13*** | sc 3 | 273.2 | 273.7 | 271.7 | 269.8 |
| Case 14*** | sc 3 | 275.7 | 275.5 | 272.5 | 271.1 |
| Case 15*** | sc 3 | 278.2 | 277.3 | 273.5 | 272.4 |

\* Simulation with Neumann BC and $H_s = 0\ K\ m\ s^{-1}$ over entire domain

\*\* **Calculated from a linear extrapolation of the potential temperature gradient between the two lowest grid-points above topography. Absolute temperature calculated with reference pressure $1000\ hPa$. The mean temperatures over land contain areas up to $70\ m$ surface elevation. In inversion conditions this results in a higher mean temperature than only for the lowest areas.**

\*\*\* **Emissions only from the largest street in the Bergen valley.**



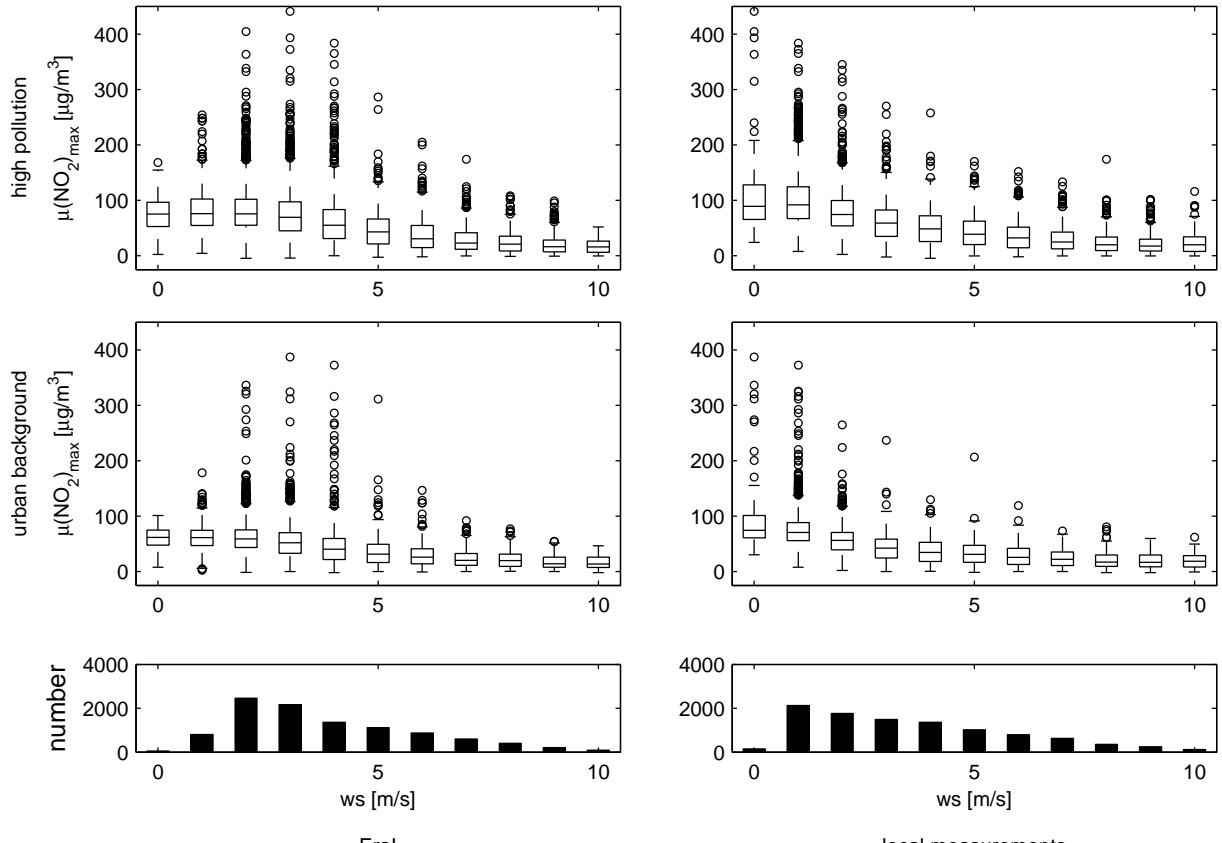

**Figure 1:** **NO$_2$ concentrations at the two air quality reference stations in Bergen against the 10 m wind speeds (ws) over the Bergen valley from EraI and from local measurements inside the valley between 2003 and 2013. EraI data are available every 3 hours. NO$_2$ concentrations and the local measured winds represent hourly means. The NO$_2$ concentrations are the 3-hourly maxima and the local wind data are the 3-hourly means around the three-hour time steps. Only wintertime data (Nov-Feb) are included. The edges of the boxes are the 25th and 75th percentiles. Lines inside the boxes show the median. The maximum whiskers length is 1.5 times the distance between the 25th and 75th percentiles or the maxima and minima of the data. Outliers with higher or lower concentration values than that are shown as circles. The lower panels show the number of simultaneously valid wind and pollution measurements within each wind speed bin.**





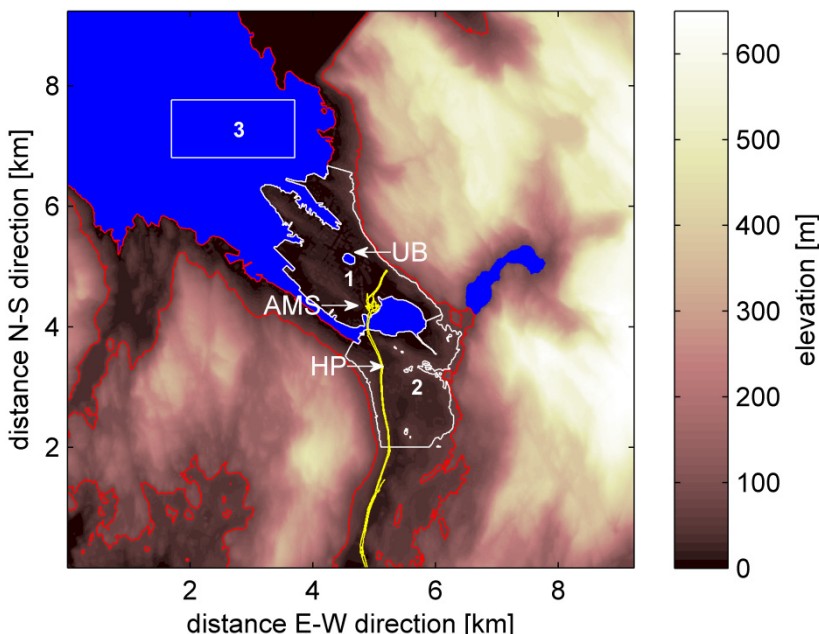

**Figure 2: Topographic map of the computational domain for the model simulations. Blue colour indicates water. The white contour line marks the areas 1 and 2 selected for averaging of the wind and passive tracer concentration. The red contour line marks the area of passive tracer emissions in most of the experiments. Both lines are overlapping along the coast. The yellow line represents the main street along the valley, eventually disappearing into a tunnel after having crossed the valley. White arrows indicate the positions of the wind measurements (AMS), and the UB and HP reference stations for NO$_2$ air pollution. The area for averaging of the 2 m temperature over the fjord in Table 1 is marked as area 3; averaging of the 2 m temperature over the city centre is done separately over areas 1 and 2.**





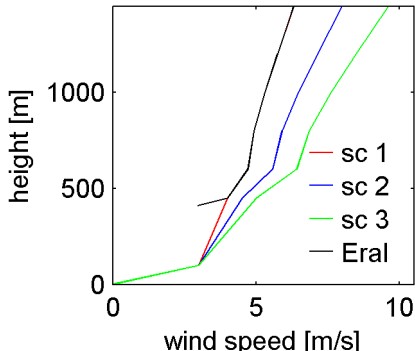

**Figure 3: Geostrophic wind profile scenarios used for the model simulations. The black line shows the mean EraI wind profile over the Bergen valley for all high pollution cases (they start only at $410\,m$ due to the low horizontal resolution of EraI). Above $450\,m$ the red and black lines are overlapping.**



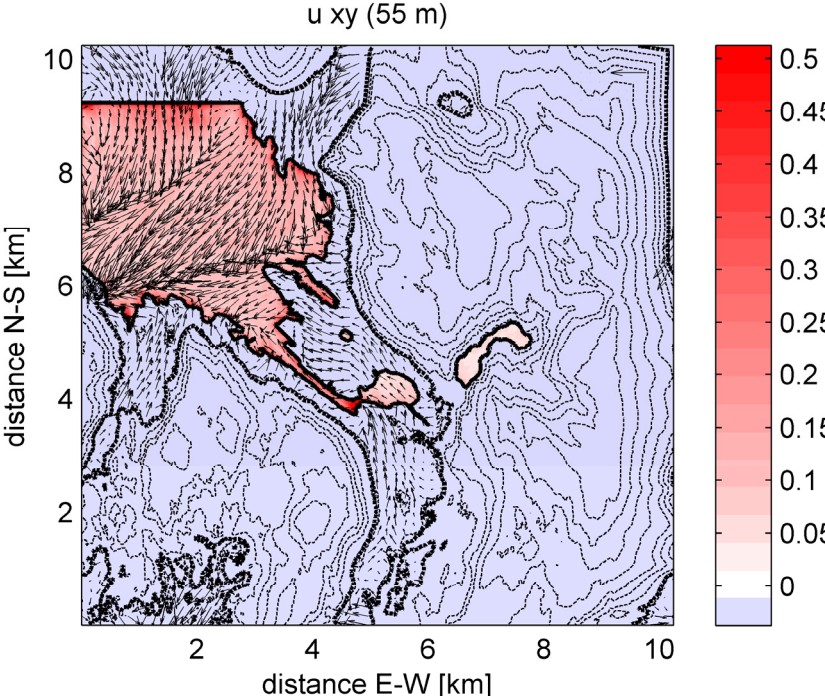

**Figure 4: The 55 $m$ wind field from case 3 together with $H_s$ [$K\,m\,s^{-1}$]. The figure shows the mean fields over the last 15 $min$ of the 12 $h$ simulation time. Wind vectors point into the flow direction. The wind-vector in the upper right corner indicates the scale of the wind vector length. It shows a wind speed of 3 $m\,s^{-1}$. The 55 $m$ topographic line is indicated by the thick black dashed line. The water-land interface is shown by the thick solid line.**





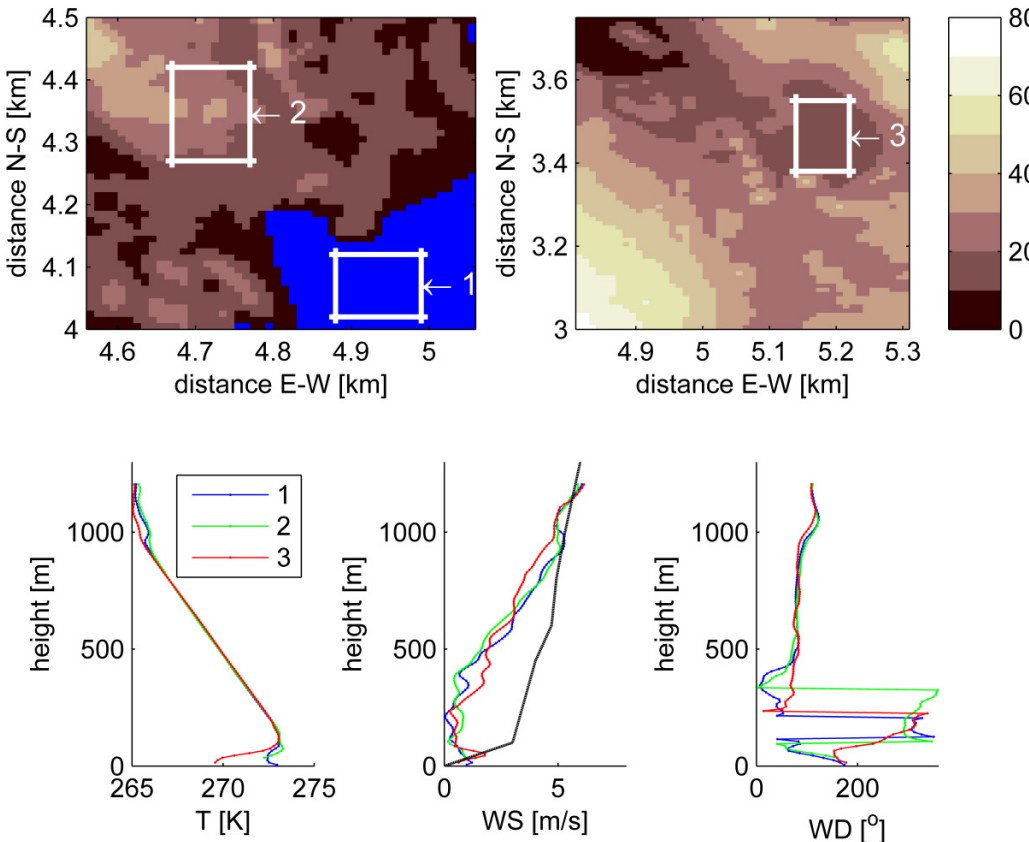

**Figure 5: Vertical profiles for temperature and wind speed and wind direction (lower panels) at three locations in the Bergen valley (marked in the upper panels). The figures show the 15** *min* **mean data corresponding to case 3 presented in Fig. 4. All profiles are horizontal averages over the areas indicated in the top panels. The blue areas indicate water surface. The brown shading gives the surface elevation in m. The AMS is located in the centre of area 2. The geostrophic wind profile for this scenario is indicated as a black line in the lower centre panel.**





**Figure 6: Wind-fields at 55 $m$ height and passive tracer concentration 2 $m$ above the surface. The top, middle and bottom panels show the results for wind speed scenario 1, 2 and 3, respectively. All data are means over the last 4 output steps of the 12 $h$ simulation. Each output step is an average of 15 $min$. Colour and wind speed scales are the**



same in all panels. Darker colour means higher tracer concentrations. The domain is cut off at the right boundary since the topography here was mostly above $55\,m$. The wind-vector in the upper right corner indicates a wind speed of $5.2\,m/s$. The water-land interface is shown by the black solid line.

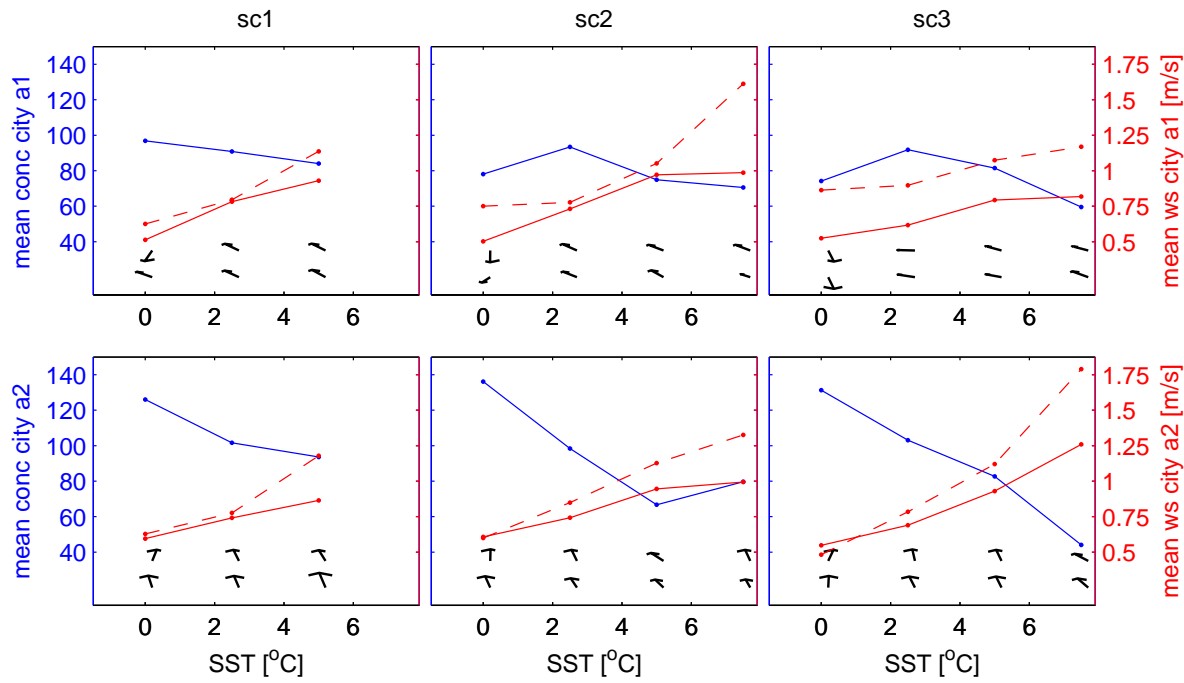

Figure 7: Mean passive tracer concentration (blue) and wind speed at $55\,m$ (red, dashed) and $10\,m$ above the surface (red, solid) for area 1 and 2 as indicated in Fig. 1 for the three wind speed scenarios as indicated in Table 1. The wind-vectors indicate the mean wind direction in the area at the $55\,m$ height level (top) and $10\,m$ above the ground (bottom). Vectors point into the direction of the flow. All data are averaged over the last 4 output steps of the $12\,h$
10   simulations corresponding to Fig. 6.





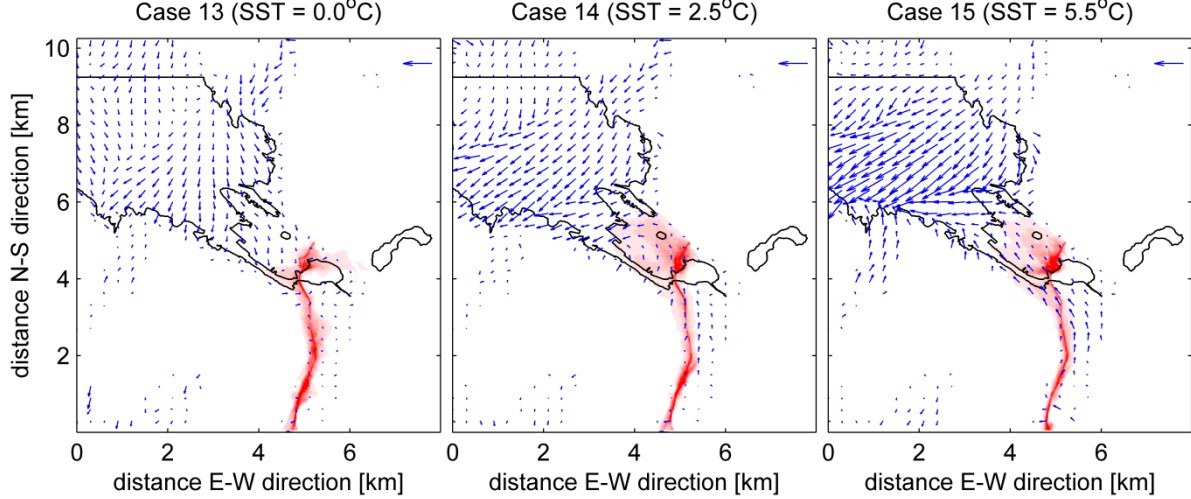

**Figure 8: Same as Fig. 6 but for the cases with emissions only from the main street inside the Bergen valley. The colour coding for the concentration is the same in all three panels, but different from Fig. 6. Shading is chosen as to show the horizontal distribution of the pollutants and not the maximum values over the street.**