# Peer review of "Sensitivity of local air quality to the interplay between small- and large-scale circulations: a Large Eddy Simulation study"

_Atmospheric Chemistry and Physics, 2016_

## Referee Comment (RC1) · Anonymous Referee #1 · 30 Nov 2016

This article is an example of a successful application of LES to explain the observed the increase of pollution concentration in the city due to the interplay between the large-scale circulation and the circulation forced by the local topographic and thermal forcing. The conclusions are confirmed by the comparison of the numerically obtained effects and the measurements. The authors showed that the physically reasonable results can be obtained even under the highly simplified setup of the numerical experiment. This result justifies the use of the standing-alone LES for post-processing of the data obtained from large-scale forecasting models to clarify the local air quality predictions over urban areas.

The following corrections should be done before the paper is published:

1) Page 5, line 30 - page 6, line 7. This paragraph can be greatly shortened or removed. There is no necessity to describe the intrinsic details of the code which are not directly connected to the physical problem.

2) The model resolution is 10 m (page 6, line 24), so the grid step is not fine enough to perform building resolved simulation. The methods of accounting of the urbanized surface should be described, at least the roughness parameter and the displacement height should be specified. Are the results sensitive to the chosen drag coefficients?

3) Page 8, lines 23-24. It might be explained what the authors mean when writing about the temperature at the level 2 m. The method of extrapolation from the rough grid should be specified.

4) Page 8, line 30. The authors wrote "The land surface temperature can therefore adapt to the fjord temperature due to the advection of relatively warmer air". It is not evident and depends on the heat fluxes balance. Moreover, the breeze circulation should transfer air from the land to the fiord (as it is shown in Fig. 4) but not vice versa.

5) Figure 5 should be redrawn, because the discontinuities in wind direction are artificial and connected with periodicity of the angle scale.

Common comments:

a) The volume of the paper could be reduced by avoiding long discussions.

b) The grid resolution sensitivity tests are not presented. It will be better to perform at least two additional runs for any presented case with finer and rougher grids to ensure the stability of the results and conclusions.

c) PALM code permits the calculations inside the non-periodic domains. It is recommended to perform at least one additional numerical experiment to show that the periodicity does not influence the results.

d) It would be better to present some turbulence statistics (at least the second mo-

ments) in ABL above the land and the sea to ensure that LES reproduces turbulent dynamics.

I am in favor of publishing this paper in Atmos. Chem. Phys. after clarifying these issues.

---

## Referee Comment (RC2) · Anonymous Referee #2 · 2 Mar 2017

Local air quality is affected by the emission and interplay between large scale forcing and local topography. This article is presenting an analysis how the local effects can interact with large scale meteorology to produce conditions with a strong stagnation in Bergen, a city located in a coastal valley. The results of the study are applicable only to the study region, and thus of limited interest. However, the usage of large eddy model is novel and shown to be very usable to study such conditions. Thus the study has potential to motivate further utilization of similar models in different locations and hopefully also in city planning. Thus I recommend this article to be published after the following remarks are considered.

1) My main concern is related to periodic boundary conditions used in the simulations.

There is some discussion how it could affect the results, and suggestion that larger domain should be used. 1000m buffer zone is used with linearly interpolated surface. Is it tested that this 1000m is enough? For example in E-W direction the slope is quite steep, if I understood the method correctly. Later there is discussion on page 11 that the artificially generated outflow at the northern border is affecting the results, but not expected to change the conclusions. This should be tested, maybe using non-periodic boundaries are larger domain.

2) Case 0 is not analyzed. It would help a non-expert reader a lot if there would be 3D figure (similar to figure 1 in Wolf et al. (2014)), where the flows in the valley would be shown schematically in case when there is no sensible heat forcing from the surface. This could be included already in section 3.1. After such a plot figures 4-7 and the changes in local circulation would be easier to understand.

3) Page 5: Lines 30 onward. I agree with Reviewer 1 that this is too detailed and could be removed.

4) Page 6: lines 13-14: Based on Figure 2 it is not so obvious that the Bergen valley is open towards south-west.

5) Figure 2: It would be nice to see the geostrophic wind direction added to figure 2, or maybe figure 4.

6) EraI wind speeds from 410m are used down to 100m. Would the results change if the information from 10m wind speeds would be used to create the initial profiles?

7) Figure 6: The colormap for the tracer concentration is quite hard to interpret. Would it be better to compare against reference simulation and present the difference, or maybe use some other colormap to make plots easier to interpret.

---

## Author Comment (AC1) · 11 May 2017

We would like to thank both referees for their help to improve this manuscript. Below is a list with point for point answers to both referees. The referee comments are repeated in italic fond, eventual comments are given in normal font. Eventual changes to the text are given in blue colour.

Referee #1

1) *Page 5, line 30 - page 6, line 7. This paragraph can be greatly shortened or removed. There is no necessity to describe the intrinsic details of the code which are not directly connected to the physical problem.*

    We agree that the paragraph is too detailed at this place.

    We have moved it from the main text into the supplementary material.

2) *The model resolution is 10 m (page 6, line 24), so the grid step is not fine enough to perform building resolved simulation. The methods of accounting of the urbanized surface should be described, at least the roughness parameter and the displacement height should be specified. Are the results sensitive to the chosen drag coefficients?*

    As a basis for the production of the topographic map we use laser scanning data. The smoothing to a 10 m resolution from the high resolution laser scans implicitly includes a surface displacement in the case of strongly variable topography. No further displacement height is included into the simulations as the non-resolved roughness elements are small compared to the resolved mountain topography (see picture below). The unresolved roughness length in LES of places embedded in complex topography is an interesting question itself but goes beyond the scope of the work presented here. We have therefore chosen a constant roughens length across the entire domain of 0.5 m. A test with roughness length of 2 m showed qualitatively similar results with some quantitative differences. We therefore assume the roughness length to be a relevant parameter that should be studied further but assume the overall conclusions of this publication not to be dependent on it.

[Figure]

    The above stated paragraph is included into the main text (page 7, line 2 in the tracked changes version).

3) *Page 8, lines 23-24. It might be explained what the authors mean when writing about the temperature at the level 2 m. The method of extrapolation from the rough grid should be specified.*

    The $2\,m$ temperature is calculated as a linear downward extrapolation of the temperature curve between the first two vertical grid-levels above the topography at 5 and 15 m.

    The above stated paragraph is included into the main text (page 8, line 26 in the tracked changes version).

4) *Page 8, line 30. The authors wrote "The land surface temperature can therefore adapt to the fjord temperature due to the advection of relatively warmer air". It is not evident and depends on the heat fluxes balance. Moreover, the breeze circulation should transfer air from the land to the fiord (as it is shown in Fig. 4) but not vice versa.*

Through the recirculation in the valley and the limited domain size, the domain mean air temperature will adapt over time to the different sea surface temperatures. After very long simulations the simulations with the same wind-speed scenario should therefore converge (when ignoring the static temperatures over the smaller water bodies aside the fjord) to the same circulation with simple constant offsets in the surface/air temperature according to the fjord surface temperature.

An improved explanation is given in the text (page 9, line 5)

5) *Figure 5 should be redrawn, because the discontinuities in wind direction are artificial and connected with periodicity of the angle scale.*

We agree with this comment.

The figure has been redrawn with dots marking the wind-direction instead of lines and a shift in the wind-direction scale in order to avoid discontinuities as much as possible (Figure 5).

a) *The volume of the paper could be reduced by avoiding long discussions.*

We have removed some discussion and moved some parts to the supplementary material.

b) *The grid resolution sensitivity tests are not presented. It will be better to perform at least two additional runs for any presented case with finer and rougher grids to ensure the stability of the results and conclusions.*

We have conducted simulations with each double and half the resolution for scenario 6.

For comparison, second moments of the turbulence statistics are included into the supplementary material (Figures S.1-S.3).

c) *PALM code permits the calculations inside the non-periodic domains. It is recommended to perform at least one additional numerical experiment to show that the periodicity*
*does not influence the results.*

We are interested in the problem of fluid dynamics inside a cavity. This could be sufficiently answered with the periodic boundary conditions. We now conducted a new set of experiments with a stronger focus on the emissions in the valley but with a larger domain (12.79 x 17.27 km$^2$). This includes an increase of the domain both on the southern and the eastern edges. The narrow artificial channel in the topographic input used for this manuscript could therefore be removed and the steep tapering zone in the east was relocated. While comparison is difficult due to some changes in the model setup, the general picture remains consistent with the simulations presented in this manuscript, especially the wind shear around the top of the inversion. The results are therefore to a degree independent of the exact choice of the domain size. While being an interesting possibility to use non-periodic boundary conditions, this project was already extremely costly in terms of computational time. We see further tests into this direction therefore as beyond the scope of this publication.

d) *It would be better to present some turbulence statistics (at least the second moments) in ABL above the land and the sea to ensure that LES reproduces turbulent dynamics.*

We agree with this comment.

Second moments in the ABL above the land and sea are added to the supplementary material (Figures S.1-S.3) and briefly discussed there.

Referee #2

1) *My main concern is related to periodic boundary conditions used in the simulations. There is some discussion how it could affect the results, and suggestion that larger domain should be used. 1000m buffer zone is used with linearly interpolated surface. Is it tested that this 1000m is enough? For example in E-W direction the slope is quite steep, if I understood the method correctly. Later there is discussion on page 11 that the artificially generated outflow at the northern border is affecting the results, but not expected to change the conclusions. This should be tested, maybe using non-periodic boundaries are larger domain.*
See answer c) to Referee #1. This is also true for differently sized buffer zones.

2) *Case 0 is not analyzed. It would help a non-expert reader a lot if there would be 3D figure (similar to figure 1 in Wolf et al. (2014)), where the flows in the valley would be shown schematically in case when there is no sensible heat forcing from the surface. This could be included already in section 3.1. After such a plot figures 4-7 and the changes in local circulation would be easier to understand.*
We included a figure showing the results from Case 0 into the manuscript. For consistence and ease of recognition the figure is however done in the same way as Fig. 4. The figure mentioned by the reviewer is a 3-D colour figure showing the local topography. Combining such a figure with information on circulation details would in our estimation be too complex.
The figure is included as the new Figure 6.

3) *Page 5: Lines 30 onward. I agree with Reviewer 1 that this is too detailed and could be removed.*
See answer 1) to Referee #1.

4) *Page 6: lines 13-14: Based on Figure 2 it is not so obvious that the Bergen valley is open towards south-west.*
Fig. 2 illustrates the area of the valley included in the PALM simulations. Therefore, the features of the Bergen valley further south are not visible in Fig. 2 and the information is given as a description in the text.

5) *Figure 2: It would be nice to see the geostrophic wind direction added to figure 2, or maybe figure 4.*
We agree with this comment.
We included an arrow indicating the geostrophic wind direction into Fig. 2.

6) *EraI wind speeds from 410m are used down to 100m. Would the results change if the information from 10m wind speeds would be used to create the initial profiles?*
The ERA-Interim wind is used as geostrophic wind. This component is treated as a forcing in PALM. Simulations with a few different setups showed some changes but consistent results. The specification of the geostrophic wind in an LES model within mountainous terrain is a to our knowledge open research question. As the resolution of ERA-Interim is low, the ground at the location of the Bergen valley is at 400 m above mean sea level as outlined in the text. A thorough evaluation of this question goes beyond the aim of the study at hand.

7) *Figure 6: The colormap for the tracer concentration is quite hard to interpret. Would it be better to compare against reference simulation and present the difference, or maybe use some other colormap to make plots easier to interpret.*
We agree with this comment.
We used a more suitable colourmap in the two figures showing passive tracer concentrations.